# Effects of short-term isolation on social behaviors in prairie voles

**Jesus E. Madrid**[‡], **Nicole M. Pranic**[‡], **Samantha Chu**, **Johanna J. D. Bergstrom**, **Rhea Singh**, **Joclin Rabinovich**, **Kaycee Arias Lopez**, **Alexander G. Ophir** *, **Katherine A. Tschida** *

Department of Psychology, Cornell University, Ithaca, NY, United States of America

‡ JEM and NMP are co-first authors on this work.
* ophir@cornell.edu (AGO); kat227@cornell.edu (KAT)

**Data Availability Statement:** All data associated with this study are made available through Cornell eCommons (https://doi.org/10.7298/dj90-fc26).

## Abstract

Social isolation affects the brain and behavior in a variety of animals, including humans. Studies in traditional laboratory rodents, including mice and rats, have supported the idea that short-term social isolation promotes affiliative social behaviors, while long-term isolation promotes anti-social behaviors, including increased aggression. Whether the effects of isolation on the social behaviors of mice and rats generalize to other rodents remains understudied. In the current study, we characterized the effects of short-term (3-days) social isolation on the social behaviors of adult prairie voles (*Microtus ochrogaster*) during same-sex and opposite-sex social interactions. Our experiments revealed that short-term isolation did not affect rates of ultrasonic vocalizations or time spent in non-aggressive social behaviors and huddling during same-sex and opposite-sex interactions. Unexpectedly, although short-term isolation also did not affect time spent in resident-initiated and mutually-initiated aggressive behavior, we found that short-term isolation increased time spent in visitor-initiated aggression during male-male interactions. Our findings highlight the importance of comparative work across species and the consideration of social context to understand the diverse ways in which social isolation can impact social behavior.

## Introduction

Social isolation is increasingly recognized as a contributor to the development of mental and physical illnesses [1–5]. For example, in humans, feelings of loneliness are correlated with increased rates of depression, anxiety, substance abuse, and decreased cognitive function [6–9]. Furthermore, social isolation promotes markers of inflammation, exacerbates cardiovascular conditions, disrupts sleep quality, and dysregulates stress response reactivity [10–13]. On the other hand, social support has been shown to mitigate the deleterious effects of physiological and psychological stressors [14, 15]. Social connectedness reduces the likelihood of depression, anxiety, and substance use disorder [16–18]. Thus, an overwhelming amount of evidence highlights the importance of our social environment to our well-being.

Humans, however, are not the only animals that are affected by their social environment. Social isolation is an impactful stressor with lasting effects on the brain and behavior across a

**Funding:** The author(s) received no specific funding for this work.

**Competing interests:** The authors have declared that no competing interests exist.

variety of mammals [19, 20]. In rodents, for example, social isolation increases levels of systemic inflammation and promotes anxiety-like and despair-like behaviors [21–23]. As in humans, social support in rodents buffers the effects of stressors, improves social investigation, and diminishes fear responses [14, 15, 24]. The consistent association between the quality (or absence) of social connections and psychophysiological health outcomes is a fundamental aspect of social animals across various taxa including primates, birds, fish, and invertebrates [20, 25].

In addition to effects on health outcomes, social isolation is well known to affect social behavior, and extensive literature has investigated the effects of long-term (> 2 weeks) social isolation on the social behaviors of traditional laboratory rodents, including mice and rats. Although a small number of studies in rats have reported that long-term isolation promotes social investigation during subsequent social interactions [26, 27], numerous studies in both rats and mice found that long-term isolation promotes anti-social behaviors, including decreasing social approach and social preference [28], increasing anxiety [29, 30], and increasing aggression [21, 31–34].

Despite the accumulating evidence that social isolation fundamentally impacts many aspects of health and wellbeing, studies have primarily focused on the effects of long-term isolation at the cost of investigating short-term periods of isolation and the consequences that follow. This lack of research is notable because most social organisms do not experience long-term isolation under natural conditions, whereas short-term isolation is sure to be common in nature and therefore more ecologically valid. A small number of studies in mice and rats have reported that short-term (< 2 weeks) isolation increases social motivation during subsequent interactions with conspecifics. Short-term isolation increases social investigation and social grooming in male rats [35], increases play behaviors in juvenile rats [36], and increases social preference in male mice [37]. Given that prior work has tended to focus on the effects of social isolation on male behavior, we recently characterized the effects of short-term isolation on the social behaviors of both female and male mice during opposite-sex and same-sex interactions [38]. We found that although short-term isolation exerted relatively subtle effects on the social behaviors of male mice when they subsequently interacted with male and female social partners, female mice exhibited robust changes in social behavior following a 3-day period of isolation. Compared to group-housed females, single-housed females spent more time investigating a novel female social partner, produced higher rates of ultrasonic vocalizations (USVs) during same-sex interactions, and engaged in same-sex mounting of novel females [38]. Studies such as these support the idea that short-term isolation can influence social behavior in mice and rats and highlight the importance of considering whether such effects vary according to sex and social context. Whether the effects of short-term isolation on the social behaviors of mice and rats generalize to less traditional rodent models remains unknown.

Although work in traditional laboratory animals has established that both long-term and short-term isolation can impact social behavior, the relevance of such work to humans may be limited because these rodents do not demonstrate persistent and selective social bonds characteristic of humans. Species with prolonged social bonds across their lives, such as the prairie vole, may be particularly affected by social isolation [24, 39–41] and therefore may be more suitable to model the effects of social isolation on human social behavior. Most prairie voles will form socially monogamous pairs and engage in biparental care [42–44]. Moreover, under natural conditions, some prairie voles disperse from the nest to establish their own territories, whereas others remain at the nest living in communal groups as adults [45]. Thus, prairie voles are likely to experience a range of periods of social isolation and communal living during their natural lives.

Indeed, research has demonstrated that social isolation from cage-mate siblings is sufficient to produce changes in prairie vole behavior and physiology similar to those observed in humans, including increased indicators of learned helplessness and anhedonia, decreased exploratory behavior, and disruption of autonomic, cardiac, and immune functions [39, 46–55]. Social isolation has also been shown to impact the social behaviors of prairie voles. For instance, social isolation can increase aggression towards same-sex conspecifics and pups [46, 51]. In social affiliation tests, socially isolated prairie voles have been found to spend more time in a social chamber, spend more time sniffing an unfamiliar conspecific, and spend less time grooming and huddling with an unfamiliar conspecific [52, 54, 56]. Notably, like work with traditional laboratory rodents, most of these studies have focused on the effects of long-term social isolation [47, 48, 52, 57, 58].

Considerably less research exists on the effects of short-term social isolation on the social behavior of prairie voles. Here, we replicate the design of our recent work in mice [38] to ask how short-term (3-day) social isolation affects the social behaviors of female and male prairie voles. We considered the effects of short-term isolation on both aggressive and non-aggressive social behaviors. We also measured the effects of short-term isolation on the production of USVs. Previous studies have found that rates and acoustic features of rodent USVs are responsive to both short- and long-term social isolation [38, 59–62], and prairie voles produce USVs during social encounters [63–65]. In addition, because voles may experience different types of social motivation during different social contexts [66], we tested the effects of short-term social isolation on these social behaviors produced during female-female interactions, male-male interactions, and male-female interactions.

## Materials and methods

### Subjects

Male and female prairie voles (*Microtus ochrogaster*) used in this experiment were F2 and F3 generation lab-born animals derived from wild-caught breeders trapped in Champaign County, Illinois, USA. Subjects born to F1 or F2 breeders were housed with parents and littermates until weaning at postnatal day (PND) 21, housed with all littermates until PND42-45, and then separated by sex and housed with their same-sex littermates until the start of the experiment (> PND60). All animals were housed in standard polycarbonate rodent cages (29 × 18 × 13 cm) lined with Sani-chip bedding and provided nesting material. We provided animals free access to water and food (Rodent Chow 5001, LabDiet, St. Louis, MO, USA). All animals were housed on a 14:10 light-dark cycle with ambient temperature maintained at 20 ± 2°C. Sex was assigned based on external genitalia. All experiments and procedures were conducted according to protocols approved by the Cornell University Institutional Animal Care and Use Committee (protocols #2020–001 and #2013–0090).

### Study design

We used a between-subjects design to measure the effects of short-term (3-day) social isolation on the social behaviors of sexually naïve, non-bonded, adult (> PND60) prairie voles during same-sex and opposite-sex interactions. Subjects were selected from cages housing at least two adult, same-sex littermates. One sibling from each cage was assigned to the single-housed condition and was separated from its littermate(s) and housed alone for three days in a clean cage with bedding and nesting material prior to being placed in a social interaction test (see below). A second sibling from each cage was assigned to the group-housed condition. Group-housed subjects were removed from their home cage and were placed directly in the social interaction

test. If a selected litter had more than two same-sex siblings, all remaining siblings from that litter were transferred to a new home cage and were not used in the experiment.

## Social interaction tests

The social interaction test began by transferring a subject (either group-housed or single-housed) within its home cage into a custom-made Plexiglas chamber (29 × 18 × 13 cm) that fit snugly around the home cage to prevent animals from escaping when the cage lid was removed. The home cage and its Plexiglas 'sleeve' were placed inside a sound-attenuating recording chamber (Med Associates) equipped with an ultrasonic microphone (Avisoft, CMPA/CM16), an infrared light source (Tendelux), and a webcam (Logitech, with the infrared filter removed to enable video recording under infrared lighting) (S1 Fig). Because these animals were tested in their home cages, we refer to subjects as "residents". Resident voles were sexually naïve at the time of the experiment and had no prior social experience outside of littermate interactions. An unrelated and unfamiliar group-housed stimulus animal (i.e., a "visitor") was then placed in the resident's home cage for 30 minutes, and video and audio recordings were made. Visitors were used across multiple trials. Three out of 13 female visitors were used across both male-female and female-female trials. No visitor was used more than 11 times within 60 days and there was always at least 1 day between uses in different trials. Visitors were individually ear-tagged prior to being used in a social interaction test for identification purposes.

We measured social behaviors (see below) during social interaction testing in three social contexts. In the first context, we tested female residents when exposed to a female visitor (FF). In the second context, we tested male residents when exposed to a male visitor (MM). In the third context, we tested male residents when exposed to a female visitor (MF). For each context, we assigned 15 subjects to serve in the group-housed condition and 15 subjects to serve in the single-housed condition. A subset of social interaction tests (n = 17) were excluded from analysis for one of the three following reasons: (1) an animal jumped on top of the Plexiglas chamber and/or onto the microphone (n = 10), (2) the recording was stopped before the 30-minute mark due to experimenter error (n = 3), or (3) the visitor identity was not recorded (n = 4). Thus, our final sample sizes for FF were n = 15 (group-housed) and n = 15 (single-housed), for MM were n = 14 (group-housed) and n = 14 (single-housed), and for MF were n = 12 (group-housed) and n = 13 (single-housed).

## Behavioral measures

**USVs.**   USVs were recorded using an Avisoft recording system (UltrasoundGate 116H, 250 kHz sample rate) and detected using custom MATLAB codes with the following parameters implemented to detect prairie vole USVs: mean frequency > 17 kHz; spectral purity > 0.3; spectral discontinuity < 1.00; minimum USV duration = 5 ms; minimum inter-syllable interval = 30 ms) [38, 67]. Because recordings were unable to distinguish whether the resident or the visitor produced a given USV, we simply counted the total number of USVs produced by a dyad within a trial. To evaluate the accuracy of our USV detection, we generated spectrograms of each detected 'putative' USV from eight representative 30-minute-long audio recordings from our dataset (n = 5800 putative USVs detected in total; from n = 4 male-male trials, n = 1 female-female trials, and n = 3 male-female trials). A trained observer rated each spectrogram as either containing a USV or not containing a USV. From this analysis, we calculated that 91.2 +/- 3.6% (in total, 5305 of 5800) of putative USVs detected by the code are true USVs, and correspondingly, we estimate a false positive rate of ~8.8%.

**Social behaviors scored from video recordings.** Trained observers, blinded to context, scored behaviors from overhead video recordings of the resident and visitor in each pair. One of the authors (N.M.P.) initially scored behavior from a subset of videos in our dataset (n = 5) that collectively contained instances of all behaviors to be scored. Three additional observers were then trained on this training dataset until their scoring reached 100% agreement with that of the trainer. Outside of the training dataset, each video was scored by only one observer, and the trainer continued to perform intermittent spot checks of scoring accuracy. A spreadsheet was used to record start and stop times for each behavior. The following behaviors were scored: (1) resident-initiated aggressive behavior, (2) visitor-initiated aggressive behavior, (3) mutually-initiated aggressive behavior, (4) resident-initiated non-aggressive behavior, (5) visitor-initiated non-aggressive behavior, (6) mutually-initiated non-aggressive behavior, and (7) huddling. Aggressive behavior included chasing (i.e., pursuit associated with fighting) and fighting (i.e., biting, boxing, or tussling). Non-aggressive behavior included sniffing, following (i.e., pursuit not associated with fighting), and grooming. For both aggressive and non-aggressive behavior, directional behavior (i.e., resident-initiated or visitor-initiated) was defined as instances in which one animal approached the other and the behaviors listed above resulted. Mutually-initiated behavior was defined as instances in which the two animals simultaneously approached each other and the behaviors above resulted. Huddling was defined as instances when the resident and visitor remained in side-by-side physical contact for more than 3 seconds without otherwise interacting. Thus, we scored aggressive behavior, non-aggressive behavior, and huddling as mutually exclusive, non-simultaneous events. No instances of mounting were observed in our dataset.

## Statistical analyses

To examine normality of residuals for the relevant data distributions, we visually inspected quantile plots of residuals. Cases in which residuals diverged notably from the 45-degree line of a normal distribution were deemed non-normally distributed and were analyzed by fitting the data to a generalized linear mixed model with a negative binomial family. We quantified huddling behavior as a binary variable (Y/N) and then analyzed it by fitting it to a generalized linear mixed model with a binomial family. We did not analyze the number of seconds animals were engaged in huddling because very few pairs engaged in any huddling behavior (n = 20/83). All models included 'visitor identity' as a random factor to control for the fact that some visitors were used across multiple trials. All p-values for pairwise comparisons were corrected for multiple comparisons using the Tukey honestly significant difference (HSD) test. Summary statistics provided in text represent mean values ± standard deviation. All statistical analyses were carried out using R 4.3.0 (R Core Team, 2023) and R Studio 2023.03.1+446 (Posit team, 2023).

## Results

### Effects of short-term isolation on prairie vole USV production in different social contexts

Rates of USVs produced by prairie vole pairs significantly differed as a function of social context (Fig 1; $X^2 = 33.14$, $p < 0.001$). However, we found no effect of resident housing condition on USV production ($X^2 = 0.87$, $p = 0.35$), and the interaction between social context and housing condition was also not significant ($X^2 = 0.31$, $p = 0.86$). Notably, MF pairs produced an average of 1174.9 ± 993.3 USVs, significantly more than both FF pairs (368.4 ± 258.7) and MM pairs (447.0 ± 306.2) ($p < 0.001$ and $p < 0.001$, respectively). These data indicate that prairie vole USV rates are influenced by social context but are not affected by short-term isolation.

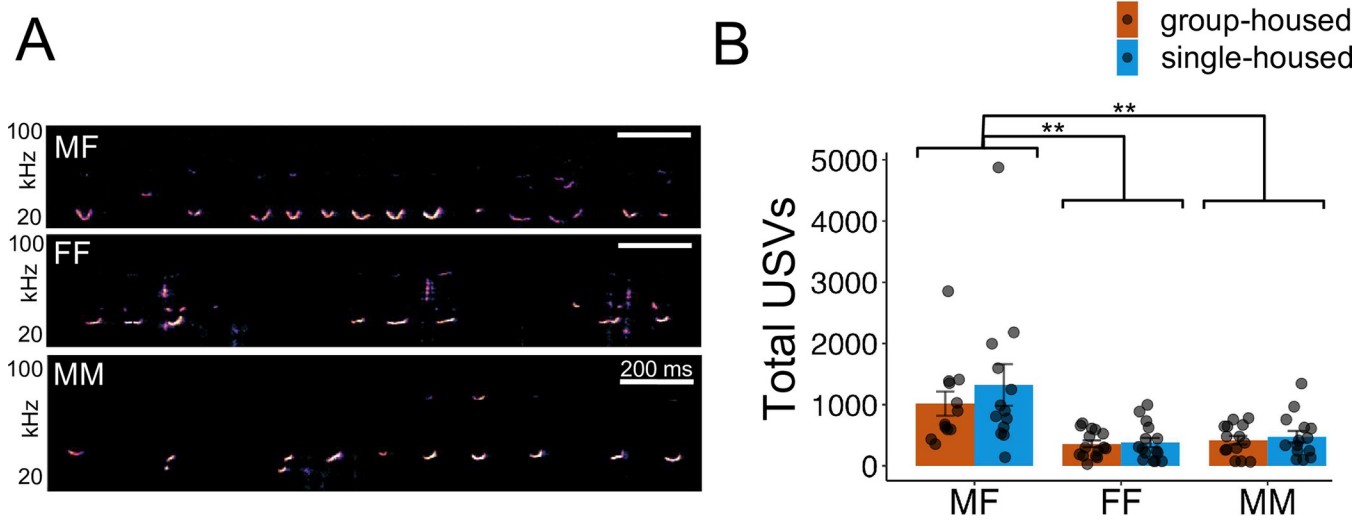

**Fig 1. Effects of short-term social isolation on prairie vole USV production in different social contexts.** (A) Spectrograms of representative USVs produced by prairie voles during opposite-sex (MF) interactions and during female-female (FF) and male-male (MM) interactions. (B) Quantification of total USVs produced during MF, FF, and MM interactions. Orange, trials with group-housed residents; blue, trials with single-housed residents. Bars indicate mean values, and error bars indicate standard errors. Double asterisks, p < 0.001.

## Effects of short-term isolation on prairie vole aggression in different social contexts

We scored aggressive behavior according to which animal (the resident, the visitor, or both) initiated the behavior. Time spent engaged in aggressive behavior initiated by the resident did not differ by resident housing condition (Fig 2A; $X^2 = 1.41$, p = 0.23) or by social context

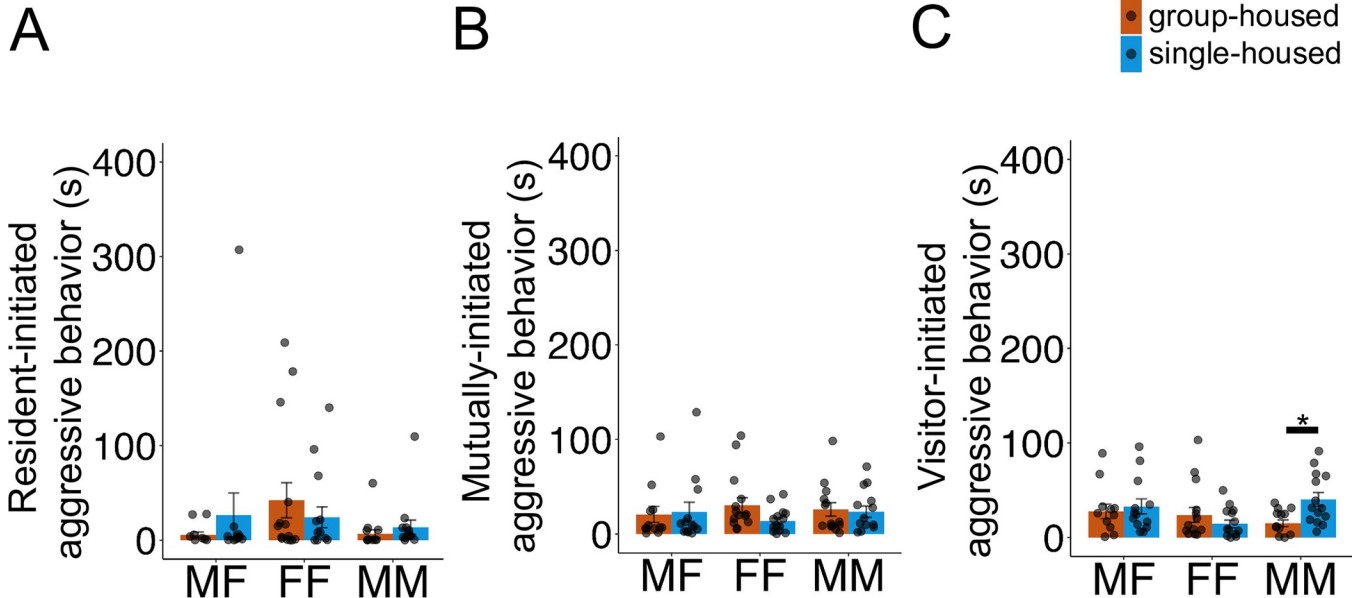

**Fig 2. Effects of short-term isolation on prairie vole aggressive behavior in different social contexts.** Time (in seconds) spent engaged in (A) resident-initiated, (B) mutually-initiated, and (C) visitor-initiated aggressive behavior during social interaction trials is shown. Orange, trials with group-housed residents; blue, trials with single-housed residents. Bars indicate mean values, and error bars indicate standard errors. Single asterisks, p < 0.05.

($X^2 = 5.14$, p = 0.08), and these factors did not show a significant interaction ($X^2 = 3.38$, p = 0.18).

Similarly, mutually-initiated aggressive behavior did not differ by resident housing condition (Fig 2B; $X^2 = 0.91$, p = 0.34) or by social context ($X^2 = 0.68$, p = 0.71), and these factors did not show a significant interaction ($X^2 = 1.99$, p = 0.37).

Unexpectedly, although visitor-initiated aggressive behavior did not show a significant main effect of resident housing condition (Fig 2C; $X^2 = 1.5$, p = 0.22) or social context ($X^2 = 1.78$, p = 0.41), the interaction between these two factors was significant ($X^2 = 6.08$, p = 0.048). Post hoc comparisons revealed that MM pairs with a single-housed resident spent more time engaged in visitor-initiated aggressive behavior than pairs with a group-housed resident (40.3 ± 26.9 s and 15.0 ± 13.4 s, respectively; p = 0.01). Furthermore, all visitor males (4 total) that were used in MM trials were tested with both single-housed residents and group-housed residents. Notably, 3 of these 4 male visitors spent more time engaged in visitor-initiated aggression during trials with single-housed residents compared to trials with group-housed residents (S2 Fig). Follow-up analyses demonstrated that both the mean number of visitor-initiated bouts of aggressive behavior and the mean duration of these bouts were significantly greater in MM trials with single-housed residents when compared to trials with group-housed residents ($X^2 = 4.7$, p = 0.03 for bout number analysis; $X^2 = 5.5$, p = 0.02 for bout duration analysis). In summary, MM pairs with single-housed residents spent more time engaged in visitor-initiated aggression compared to MM pairs with group-housed residents. These data reveal a sex- and context-dependent effect of short-term social isolation on visitor-initiated aggressive behavior.

## Effects of short-term isolation on prairie vole non-aggressive social behavior in different social contexts

We next considered the effects of short-term isolation on prairie vole non-aggressive social behavior by categorizing periods of non-aggressive behavior as resident-initiated, mutual, or visitor-initiated. Time spent in resident-initiated non-aggressive social behavior showed a significant main effect of social context (Fig 3A; $X^2 = 8.90$, p = 0.01), but there was no significant main effect of resident housing condition ($X^2 = 0.52$, p = 0.47), and there was no significant interaction between resident housing condition and social context ($X^2 = 3.01$, p = 0.22). Post hoc comparisons among prairie vole dyads showed that MM pairs (26.1 ± 54.6 s) spent significantly less time engaged in resident-initiated non-aggressive social behaviors than MF pairs (80.1 ± 85.9 s; p = 0.03) and FF pairs (77.2 ± 157.7 s; p = 0.02).

Mutually-initiated non-aggressive social behavior also showed a main effect of social context (Fig 3B; $X^2 = 6.89$, p = 0.03) but no main effect of resident housing condition ($X^2 = 1.75$, p = 0.19) and no significant interaction ($X^2 = 3.69$, p = 0.16). Post hoc comparisons showed that MM pairs spent less time engaged in mutually-initiated non-aggressive social behavior than FF pairs (37.1 ± 43.2 and 26.5 ± 86.2 s, respectively; p = 0.049).

Finally, visitor-initiated non-aggressive social behavior showed the same pattern, where we found a significant main effect of social context (Fig 3C; $X^2 = 12.25$, p = 0.002) but no significant main effect of resident housing condition ($X^2 = 1.05$, p = 0.30) and no significant interaction between resident housing condition and social context ($X^2 = 0.24$, p = 0.88). Post hoc comparisons showed that MM pairs (30.7 ± 45.2 s) spent significantly less time engaged in visitor-initiated non-aggressive social behaviors than MF pairs (122.7 ± 96.7 s; p = 0.02) and FF pairs (85.1 ± 133.8 s; p = 0.03). Taken together, these results indicate that prairie vole non-aggressive social behavior is influenced by social context but is not impacted by short-term isolation.

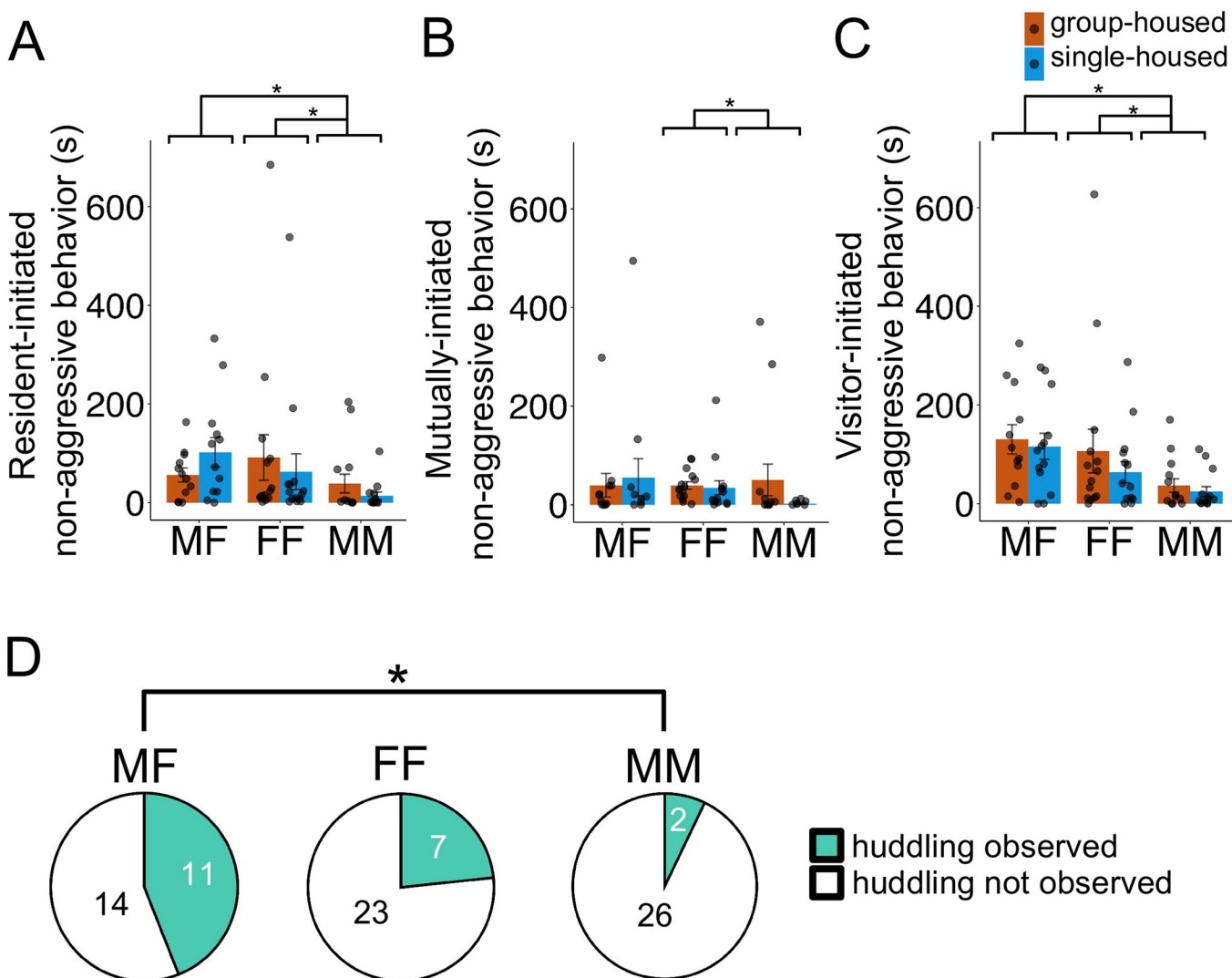

**Fig 3. Effects of short-term isolation on prairie vole non-aggressive social behavior and huddling in different social contexts.** (A-C) Time (in seconds) spent engaged in (A) resident-initiated, (B) mutually-initiated, and (C) visitor-initiated non-aggressive social behavior during social interaction trials is shown. Orange, trials with group-housed residents; blue, trials with single-housed residents. Bars indicate mean values, and error bars indicate standard errors. Single asterisks, p < 0.05. Double asterisks, p < 0.001. (D) Pie charts show the number of pairs engaged in huddling in MF (left), FF (middle), and MM (right) social interactions. White shading indicates proportion of trials in which pairs did not huddle, and teal shading indicates proportion of trials in which pairs engaged in huddling.

## Effects of short-term social isolation on prairie vole huddling in different social contexts

Although huddling (i.e., side-by-side contact) was rare among resident-visitor pairs (20 of 83 pairs engaged in huddling behavior; see S3 Fig), our results showed a significant main effect of social context on the number of pairs that engaged in huddling (Fig 3D; $X^2$ = 6.04, p = 0.049). However, we did not find a significant main effect of resident housing condition ($X^2$ = 0.31, p = 0.58) or a significant interaction between resident housing condition and social context ($X^2$ = 0.27, p = 0.87). Post hoc comparisons showed that huddling was observed more frequently in MF pairs than in MM pairs (11 of 25 MF pairs huddled and 2 of 28 MM pairs huddled; p = 0.047). As with non-aggressive social behavior, our data indicate that frequency of huddling differs by social context but is not affected by short-term isolation.

## Discussion

In this study, we measured the effects of short-term social isolation on social behaviors in prairie voles, during same-sex and opposite-sex social interactions. Surprisingly, we found that three days of social isolation did not impact rates of USV production, time spent engaged in non-aggressive social behaviors, or frequency of huddling (Figs 1 and 3). Similarly, we found no main effect of social isolation or social context on time spent in aggressive behavior (Fig 2). However, we did find that short-term isolation increased visitor-initiated aggression in a context-dependent manner, where social isolation did not impact MF or FF dyads, but MM pairs with a single-housed resident spent more time engaged in visitor-initiated aggression than MM pairs with a group-housed resident (Fig 2C). On the other hand, social context of the dyads (MF, FF, or MM) strongly influenced rates of USV production, non-aggressive behavior, and huddling. MF pairs produced higher rates of USVs than same-sex pairs (Fig 1B), MF and FF pairs spent significantly more time engaged in non-aggressive behaviors compared to MM pairs (Fig 3A and 3C), and significantly more MF pairs engaged in huddling than MM pairs (Fig 3D).

Previous work in mice and rats has demonstrated that short-term (< 2 weeks) social isolation promotes a variety of social behaviors, including social play [36], social investigation [37, 38], USV production [38], mounting [35, 38], and grooming [35]. To our knowledge, the current study is the first to consider the effects of short-term isolation on the social behaviors of freely interacting, adult prairie voles. In contrast to work in mice and rats, most behaviors that we measured (USVs, non-aggressive behaviors, and huddling) were surprisingly not affected by short-term isolation. Similarly, Sailer et al. (2022) found that juvenile prairie voles that experienced 9 days of social isolation engaged in social approach toward and social investigation of a conspecific behind a physical barrier at rates that were no different from animals that were not isolated. It is worth noting that combining social isolation while also exposing animals to a social stress regimen (i.e., chronic social defeat), however, reduced these behaviors [24]. Taken together, these studies suggest that prairie voles might not be as sensitive to short-term social isolation as mice or rats. This difference highlights the importance of comparative studies across species for understanding how social isolation impacts behavior.

Although we found that short-term isolation exerted relatively few effects on prairie vole social behavior, aggressive interactions among our dyads were influenced by social isolation; however, this effect depended on social context. Unexpectedly, we found that 3 days of social isolation promoted prairie vole aggression, but only among male-male dyads. Moreover, these behavioral differences were only observed in the visitors' aggression. How might single-housing of the resident male increase aggressive behavior of the visitor male? One possibility is that although single-housed male residents did not initiate more fights, perhaps they either promoted or prolonged altercations or perhaps they were less effective at defusing visitor-initiated aggression compared to group-housed male residents. These ideas are aligned with the hypothesis that social isolation results in a deficiency of social skills, although this hypothesis has mainly been discussed in the context of early-life social isolation and subsequent overexpression of aggression or altered courtship behaviors later in life [59, 68]. A second more practical possibility is that visitor males responded with aggression towards behaviors exhibited by single-housed resident males, but we did not capture these behaviors in our video analysis, either because they were more subtle behaviors than those that we scored (for example, postural differences) or because they could not be captured by video recordings (for example, differences in chemical signaling). Unfortunately, we are unable to distinguish between these possibilities based on the data we collected. Nevertheless, our data complement previous work showing that longer-term isolation increases aggressive behavior in female prairie voles [46, 51].

Different social contexts created by the dyads that we established impacted non-aggressive behaviors and the rates of USVs. However, the contexts impacted these behaviors differently. In many ways, the social context among dyads led to predictable outcomes. For instance, both MF and FF pairs spent more time engaged in non-aggressive behaviors than MM pairs (Fig 3A and 3C), a result that is likely best explained by intersexual competition among males. Moreover, MF pairs spent the most time engaged in social behaviors that are typically considered prosocial. Indeed, male-female pairs produced higher rates of USVs than same-sex pairs (Fig 1B) and were more likely to exhibit huddling than male-male pairs (Fig 3D). Presumably these differences in behavior are related to the reproductive context that tends to follow male-female pairings. Notably, the finding that MF pairs produced the overall highest rates of USVs aligns with previous work, which found that MF pairs of prairie voles emit higher rates of USVs than MM pairs [64, 65], and that USV production during opposite-sex interactions accompanies social investigation, mounting, and intromission [63]. Previous studies in prairie voles have manually categorized USV types (based on spectrographic shape) produced during same-sex interactions [64], described the acoustic features of USVs produced during opposite-sex interactions [63], and reported that the acoustic features of vole USVs covary with heart rate [69]. Whether the acoustic features of prairie vole USVs differ according to social context or are influenced by social isolation remains an important topic for future study.

Recent work has put forth the idea of social homeostasis, in which individuals detect the quality and quantity of social interaction, compare it to a "set-point" of optimal social contact, and then modify their social seeking behaviors to achieve optimal social contact. Specifically, the social homeostasis hypothesis proposes that animals increase their rates of prosocial behaviors following short periods of social isolation, but that long-term isolation alters the social set-point, which in turn makes social contact a negative valence stimulus and causes individuals to increase their rates of anti-social behaviors, including aggression [70]. Notably, the patterns of social behaviors that we report are inconsistent with the predictions of this hypothesis. Indeed, none of the prosocial behaviors we measured were impacted by short-term social isolation. In contrast, we found that short-term social isolation leads to a social context-dependent increase in visitor-initiated aggressive behavior. Given that most studies in prairie voles have focused on prolonged periods (> 4 weeks) of social isolation, additional work is needed to understand the time course over which aggressive behaviors emerge in single-housed females and males. Similarly, it would be interesting to know whether there are conditions under which short-term isolation enhances prosocial behaviors in prairie voles. For example, group-housed and single-housed resident prairie voles in the current study were given social interactions with unfamiliar visitors. Yet it is possible that the effects of short-term isolation on vole social behaviors might differ if subsequent interactions were conducted with familiar voles. This distinction in the familiarity of the visitor may be relevant considering that exposure to familiar and unfamiliar conspecifics (or even unfamiliar heterospecifics) elicits different patterns of neuronal activation throughout the 'social behavior network' in the prairie vole brain [71].

In summary, our study highlights the paramount importance of considering factors that can influence whether and how isolation affects social behavior. In particular, the duration and timing of isolation, the natural history of the species studied, and the sex of the social interactants are all critical factors that merit consideration. Doing so will lead to a deeper understanding of the various ways in which social isolation impacts social behavior and mental and physical health.

## Supporting information

**S1 Fig. Behavioral chamber setup.** (A) The home cage of the resident vole (a) was placed inside a plexiglass sleeve (b), with a small amount of foam padding (c) placed to fill any gaps

between the edge of the home cage and the wall of the sleeve. The chamber was equipped with an ultrasonic microphone (d) and a webcam (e). Please note that although only the resident vole is present in this image, both a resident and visitor vole were placed in the chamber for each social interaction test. (B) View of the behavioral chamber from the overhead webcam. (PDF)

**S2 Fig. Mean time spent in visitor-initiated aggressive behavior for male visitors during interactions with single-housed vs. group-housed male residents.** Mean time (in seconds) that male visitors engaged in visitor-initiated aggressive behavior when interacting with group-housed vs. single-housed male residents. Lines and data points are color-coded by the identity of the male visitor. Data points show mean values, and error bars indicate standard errors. Visitor males #4 and #5 were each used in n = 2 trials with group-housed (GH) residents and n = 2 trials with single-housed (SH) residents. Visitor male #7 was used in n = 5 trials with GH residents and n = 6 trials with SH residents. Visitor male #8 was used in n = 5 trials with GH residents and n = 4 trials with SH residents.
(PDF)

**S3 Fig. Additional quantification of huddling.** (A) Time (in seconds) that pairs of voles spent engaged in huddling during social interaction trials is shown. Orange, trials with group-housed residents; blue, trials with single-housed residents. Bars indicate mean values, and error bars indicate standard errors. (B) Pie charts showing the number of pairs engaged in huddling in MF (top), FF (middle), and MM (bottom) social interactions, shown separately for pairs that included group-housed residents vs. single-housed residents. White shading indicates the proportion of trials in which pairs of voles did not huddle, and teal shading indicates the proportion of trials in which pairs engaged in huddling.
(PDF)

## Acknowledgments

We thank Frank Drake and other CARE staff for their excellent animal husbandry. We also thank Stephen Parry from the Cornell Statistical Consulting Unit for statistical consultation.

## Author Contributions

**Conceptualization:** Jesus E. Madrid, Nicole M. Pranic, Samantha Chu, Johanna J. D. Bergstrom, Alexander G. Ophir, Katherine A. Tschida.

**Data curation:** Nicole M. Pranic, Katherine A. Tschida.

**Formal analysis:** Jesus E. Madrid, Nicole M. Pranic, Samantha Chu, Johanna J. D. Bergstrom, Rhea Singh, Joclin Rabinovich, Kaycee Arias Lopez.

**Funding acquisition:** Alexander G. Ophir, Katherine A. Tschida.

**Investigation:** Jesus E. Madrid, Nicole M. Pranic, Samantha Chu, Johanna J. D. Bergstrom.

**Project administration:** Alexander G. Ophir, Katherine A. Tschida.

**Resources:** Alexander G. Ophir, Katherine A. Tschida.

**Supervision:** Jesus E. Madrid, Nicole M. Pranic, Alexander G. Ophir, Katherine A. Tschida.

**Visualization:** Nicole M. Pranic.

**Writing – original draft:** Jesus E. Madrid, Nicole M. Pranic, Alexander G. Ophir, Katherine A. Tschida.

**Writing – review & editing:** Jesus E. Madrid, Nicole M. Pranic, Samantha Chu, Johanna J. D. Bergstrom, Rhea Singh, Joclin Rabinovich, Kaycee Arias Lopez, Alexander G. Ophir, Katherine A. Tschida.

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
