## [Decision Letter · Decision Letter 0]

4 Sep 2024

PONE-D-24-33108Effects of short-term isolation on vocal and non-vocal social behaviors in prairie volesPLOS ONE

Dear Dr. Tschida,

Thank you for submitting your manuscript to PLOS ONE. After careful consideration, we feel that it has merit but does not fully meet PLOS ONE’s publication criteria as it currently stands. Therefore, we invite you to submit a revised version of the manuscript that addresses the points raised during the review process.

Please note that one reviewer in particular had major concerns on the way some of the results are interpreted.

We look forward to receiving your revised manuscript.

Kind regards,

Luca Nelli, PhD

Academic Editor

PLOS ONE

Journal requirements: 1. When submitting your revision, we need you to address these additional requirements. Please ensure that your manuscript meets PLOS ONE's style requirements, including those for file naming. The PLOS ONE style templates can be found at https://journals.plos.org/plosone/s/file?id=wjVg/PLOSOne_formatting_sample_main_body.pdf and https://journals.plos.org/plosone/s/file?id=ba62/PLOSOne_formatting_sample_title_authors_affiliations.pdf. 2. When completing the data availability statement of the submission form, you indicated that you will make your data available on acceptance. We strongly recommend all authors decide on a data sharing plan before acceptance, as the process can be lengthy and hold up publication timelines. Please note that, though access restrictions are acceptable now, your entire data will need to be made freely accessible if your manuscript is accepted for publication. This policy applies to all data except where public deposition would breach compliance with the protocol approved by your research ethics board. If you are unable to adhere to our open data policy, please kindly revise your statement to explain your reasoning and we will seek the editor's input on an exemption. Please be assured that, once you have provided your new statement, the assessment of your exemption will not hold up the peer review process.

Reviewers' comments:

Reviewer's Responses to Questions

**Comments to the Author**

1. Is the manuscript technically sound, and do the data support the conclusions?

Reviewer #1: Partly

Reviewer #2: Partly

2. Has the statistical analysis been performed appropriately and rigorously? 

Reviewer #1: Yes

Reviewer #2: I Don't Know

3. Have the authors made all data underlying the findings in their manuscript fully available?

Reviewer #1: Yes

Reviewer #2: Yes

4. Is the manuscript presented in an intelligible fashion and written in standard English?

Reviewer #1: Yes

Reviewer #2: Yes

5. Review Comments to the Author

Reviewer #1: The paper “Effects of short-term isolation on vocal and non-vocal social behaviors in prairie voles” is the first direct assessment of the role of short-term (here, 3 days) social isolation on prairie vole social behavior. The authors assessed behavior of both males and females, and across social contexts (MM, MF, and FF). The authors find no difference in vocal behavior as a function of housing group (isolated or not), but found, akin to other rodent models, that more USVs are emitted by MF pairs than same-sex pairs. They also find no changes in affiliative behaviors as a function of housing group. In the case of aggressive behaviors though, they find an interaction between housing group and social contexts. However, as explained further below, the reporting of this result is somewhat misleading and should be reworked to more accurately frame the finding. Overall though, the findings add significantly to the literature of the effects of social isolation, and I did find it noteworthy that they do not provide evidence for the social homeostasis hypothesis in the prairie vole.

Major concerns:

1) As alluded to above, the biggest issue here is that the interpretation of the results as stated in the Abstract is misleading given the data provided. The paragraph starting at 242 tells us that ‘visitor-initiated aggression’ differed, but the interpretation of that result is that “the effects of short-term isolation on prairie vole aggression are sex- and context-dependent.” As shown in Figures 2A and B, the aggressive behavior of the isolated animals isn’t changing if they are group housed. Nor are there sex differences in the results (neither male nor female residents change their behavior in any of the contexts, nor is there an interaction within the resident-initiated or mutual-initiated behaviors). The only thing that changes is the behavioral response by the visitor. Yes, the authors provide greater context for this interpretation in the discussion, but its presence without context in the Abstract is misleading.

2) In the title and throughout the paper, the authors use the term “non-vocal social behaviors”, which I think is meant to indicate that they are social behaviors beyond USV emission. However, this reads as though they are silent social behaviors, which is not in any way assessed here. This terminology should be updated to be less ambiguous.

3) All plots need standard deviation indicators.

4) I’m not sure what data is actually presented in Figure 2. The axes say we’re looking specifically at fights, which represent a single aggressive behavior listed in the methods. But the figure caption says it’s more general “aggressive behavior” being shown. Please update this figure/caption to make it more clear what we’re looking at.

5) Expand the section for behavioral extraction.

a. How were behaviors actually scored? Was a specific program used? Did observers record start and stop times, or did they just have a stopwatch running to get total times?

b. Provide a specific definition for each behavior. This would answer several questions about the data; how was a fight specified as ‘mutually-initiated’? How did the aggressive ‘chase’ differ from the non-aggressive ‘follow’, etc.

6) Provide more information about USV extraction. The reference in the text is to a paper that uses Holy lab software to extract mouse USVs, but the parameters used in this manuscript differ. So how were the parameters optimized for prairie vole vocalizations? And if this is the first use of the code for prairie voles (if not, please provide a reference), how was the data ground-truthed and to what threshold of accuracy?

7) The discussion talks about previous work in other rodents where short-term isolation is less than 2 weeks. However, the authors chose to only do a 3 day isolation. Please explain that choice of duration. If the mouse/rat work is generally a 10 day isolation, for instance, it seems improper to compare the 3 day isolation here and claim a species difference.

8) Using total time in behavior doesn’t give a complete picture of the behavior. It would be nice to also see the raw numbers of behaviors as well as the average durations of behaviors. That would give the reader a better idea of what exactly may be changing (or not changing).

Minor concerns:

1) For the behavioral scoring, was any sort of inter-rater reliability metric used for training or for scoring the videos used in the experiment? It would be nice to have evidence of consistency in behavioral scoring.

2) For supplemental figure 1, in addition to adding the standard deviation bars, please indicate how many times each stimulus male was used within each housing type.

3) Can the authors explain the use of Tukey’s HSD for your post-hoc analysis? This test is not very conservative, and thus seems unlikely to control for spurious significance in the results.

4) In the discussion, the authors posit that (line 335), “perhaps [single-housed males] prolonged altercations.” While the data you provide is shown as total time, I assume it can be broken down to characterize the lengths of individual fights or aggressive behaviors? Can this be used to provide evidence for/against this possibility?

5) Figure 3D: the “MF>MM” seems to just be floating here. That could easily be interpreted that non-huddling is greater in MF than in MM. I would suggest finding another way to represent this result or at least describe this in the figure caption.

Reviewer #2: The goal of the research is to examine the effects of short-term social isolation on

social interaction of introduced and unfamiliar male-male, male-female and male-female adult prairie voles. The general concept is to emphasize short-term isolation in less traditional species (compared to house mice and rats) with more selective bonds such as might occur in monogamous species like prairie voles (although the effect of this is more likely to occur in familiar individuals). This study is useful for characterizing effects of isolation specifically in prairie voles because there is evidence that they may respond differently to short term isolation compared rats and house mice. It is assumed that the effects of isolation on social behavior have not been examined in other nontraditional species?

Lines 111-112: For the phrase “Previous studies have found that rates of rodent USVs

112 are responsive to both short- and long-term social isolation [38,59–62]” please provide the direction of the effect on USVs (increase vs decrease).

The “visitors” used to intrude on the resident’s home cage were used a number of times such that “no visitor was used more than 11 times within 60 days.” This seems like a large number and it is possible that the experience of the visitors could have influenced the behavior of the residents (as occurred in the current study). Identity of intruders was controlled for, but not the number of times that visitors/intruders were used. The level of stress could vary and perceived defeat could stress the visitor/intruder. Please address this issue.

A visual of the setup for recording the vocalizations would be useful, including the plexiglas sleeve and the placement of the microphone. Please also state the number of animals that were excluded for each of the three reasons stated on 173-175

In the results it was found that short term isolation of males induced the visitor males to be more aggressive. One reasonable speculative explanation is that isolated males were producing more alarm pheromones. One practical explanation emphasized was that the behaviors were likely missed. Please explain whether this could have occurred because of behavioral coding or because the entire arena/cage was not visible when behavior was recorded. Expanding on the idea that a short term isolation could decrease social skills could also be explained in more detail.

6. PLOS authors have the option to publish the peer review history of their article (what does this mean?). If published, this will include your full peer review and any attached files.

Reviewer #1: No

Reviewer #2: No

---

## [Author Response · Author response to Decision Letter 0]

26 Sep 2024

We thank the Reviewers and the Editor for their thoughtful and helpful comments. We believe that our manuscript has improved after addressing these concerns. Below, we address in turn the specific concerns raised by each Reviewer. Please note that all line numbers referenced below refer to the version of our revised manuscript that includes tracked changes, viewed as ‘all markup’.

Responses to Reviewer 1

Reviewer 1 Major Concerns

1) As alluded to above, the biggest issue here is that the interpretation of the results as stated in the Abstract is misleading given the data provided. The paragraph starting at 242 tells us that ‘visitor-initiated aggression’ differed, but the interpretation of that result is that “the effects of short-term isolation on prairie vole aggression are sex- and context-dependent.” As shown in Figures 2A and B, the aggressive behavior of the isolated animals isn’t changing if they are group housed. Nor are there sex differences in the results (neither male nor female residents change their behavior in any of the contexts, nor is there an interaction within the resident-initiated or mutual-initiated behaviors). The only thing that changes is the behavioral response by the visitor. Yes, the authors provide greater context for this interpretation in the discussion, but its presence without context in the Abstract is misleading.

We agree with this comment and have edited the corresponding sentences in the Abstract to more accurately communicate our findings (lines 25-28).

2) In the title and throughout the paper, the authors use the term “non-vocal social behaviors”, which I think is meant to indicate that they are social behaviors beyond USV emission. However, this reads as though they are silent social behaviors, which is not in any way assessed here. This terminology should be updated to be less ambiguous.

 The term “non-vocal social behaviors” was intended to indicate social behaviors beyond USV emission (as stated by the Reviewer). More generally, we used the term “non-vocal” to refer to behaviors that were scored from video recordings, as compared to USVs, which were detected from audio recordings. We agree with the Reviewer that we did not assess whether social behaviors scored from video recordings are silent, and we intend to compare the timing (and acoustic features) of USVs to the timing of social behaviors scored from video recordings in a follow-up manuscript.

For the purposes of the current manuscript, we have eliminated the terms “non-vocal” and “vocal” throughout the manuscript. The term “social behavior” is now used to encompass USVs, as well as social behaviors scored from video recordings. USVs are now defined as such (rather than referred to as “vocal” behavior), and categories of social behaviors scored from videos are listed individually where discussed. We have also updated the title of the manuscript accordingly. 

3) All plots need standard deviation indicators.

We have added the requested error bars, although we have opted to use standard error, because the addition of standard deviation error bars made it difficult to visualize the individual data points shown within the plots.

4) I’m not sure what data is actually presented in Figure 2. The axes say we’re looking specifically at fights, which represent a single aggressive behavior listed in the methods. But the figure caption says it’s more general “aggressive behavior” being shown. Please update this figure/caption to make it more clear what we’re looking at.

Thank you for catching this oversight. The y axes in Figure 2 should have been labeled “aggressive behavior” (sum of time spent in chasing and fighting; more information on chasing is provided below in response to the Reviewer’s next point). We have changed the y axis labels in Figure 2 accordingly. 

5) Expand the section for behavioral extraction.

a. How were behaviors actually scored? Was a specific program used? Did observers record start and stop times, or did they just have a stopwatch running to get total times?

Trained observers watched videos (using Windows Media Player or similar software) and used an Excel spreadsheet to record start and stop times for different behaviors. This information has been added to the Methods (lines 209-210). 

b. Provide a specific definition for each behavior. This would answer several questions about the data; how was a fight specified as ‘mutually-initiated’? How did the aggressive ‘chase’ differ from the non-aggressive ‘follow’, etc.

For both aggressive and non-aggressive behaviors, mutually-initiated behavior was defined as behavior that followed simultaneous approach by both voles. Directional behavior (either resident-initiated or visitor-initiated) was defined as behavior that occurred after one vole approached the other). 

Follows were defined as pursuit that was not associated with fighting (biting, boxing, or tussling). Chases were defined as pursuit that was associated with fighting. This information has been added to the Methods (lines 213-221).

6) Provide more information about USV extraction. The reference in the text is to a paper that uses Holy lab software to extract mouse USVs, but the parameters used in this manuscript differ. So how were the parameters optimized for prairie vole vocalizations? And if this is the first use of the code for prairie voles (if not, please provide a reference), how was the data ground-truthed and to what threshold of accuracy?

Yes, this manuscript is the first use of our USV detection code for prairie voles. The values of two of the three parameters (mean spectral purity < 0.3; mean spectral discontinuity < 1.00) do not differ from those that we use currently to detect mouse USVs. A small comment is although we used a lower spectral discontinuity threshold to detect USVs in earlier mouse studies (< 0.85; Tschida et al., 2019; Zhao et al., 2021), we later increased the value of this threshold to improve detection accuracy given the recording conditions in our vivarium space. For the third parameter (mean frequency), we lowered the threshold relative to what we use for mouse USV recordings (from 45 Hz to 17 kHz), to account for the fact that many prairie vole USVs are lower in frequency compared to mouse USVs. 

To evaluate the accuracy of our USV detection, we generated spectrograms of each detected ‘putative’ USV from eight, representative 30-minute-long audio recordings from our dataset (n = 5800 putative USVs in total; from n = 4 male-male trials, n = 1 female-female trials, and n = 3 male-female trials). A trained observer rated each spectrogram as either containing a USV or not containing a USV. From this analysis, we calculated that 91.2 +/- 3.6% (in total, 5305 of 5800) of putative USVs detected by the code are true USVs, and correspondingly, we estimate a false positive rate of ~8.8%. This information has been added to the Methods (lines 194-200).

7) The discussion talks about previous work in other rodents where short-term isolation is less than 2 weeks. However, the authors chose to only do a 3 day isolation. Please explain that choice of duration. If the mouse/rat work is generally a 10 day isolation, for instance, it seems improper to compare the 3 day isolation here and claim a species difference.

The effects of short-term isolation on behavior are understudied relative to the effects of long-term isolation, and as such, there isn’t an agreed upon standard duration of short-term isolation that is used across rodent studies. Our choice of duration was motivated specifically by our aim to replicate the design of our recent study in mice, which represents the most comprehensive characterization of the effects of short-term isolation on adult, mouse social behavior conducted to date (Zhao et al., 2021). In that study, we found robust effects of 3-days of isolation on the behavior of mice, particularly in female mice that subsequently engaged in same-sex interactions. We have modified the final paragraph of the Introduction to clarify this motivation for our choice of 3-days of social isolation (line 112).

8) Using total time in behavior doesn’t give a complete picture of the behavior. It would be nice to also see the raw numbers of behaviors as well as the average durations of behaviors. That would give the reader a better idea of what exactly may be changing (or not changing).

As noted above by the Reviewer, the main finding of our study is that the social behaviors of subject/resident prairie voles were not altered by 3 days of social isolation. As such, we do not think it is of interest to consider numbers of bouts or mean durations of bouts of these behaviors, although we do include the behavior scoring (as Excel sheets) for each trial in our dataset within the publicly accessible data repository for this study.

In the case of the one category of social behavior that was significantly altered following 3 days of social isolation (increased visitor-initiated aggressive behavior in male-male pairs that contained a single-housed resident vs. group-housed resident), we have analyzed the total numbers of bouts of visitor-initiated aggressive behaviors, as well as the mean durations of these bouts. Both increase significantly (chi-squared = 4.7, p = 0.03 for bout number analysis; chi-squared = 5.5, p = .02 for bout duration analysis). We include plots of these analyses below for the Reviewer, and these findings have also been added in text to the Results section (lines 275-279). 

Reviewer 1 Minor concerns

1) For the behavioral scoring, was any sort of inter-rater reliability metric used for training or for scoring the videos used in the experiment? It would be nice to have evidence of consistency in behavioral scoring.

One of the authors (N.M.P.) initially scored behavior from a subset of videos in our dataset (n = 5) that collectively contained all behaviors to be scored. Three additional observers were then trained on this training dataset, and they scored additional videos only after their scoring matched the trainer’s scoring perfectly for each video (100% inter-observer agreement). Outside of the training dataset, each video was scored by only one observer, and the trainer continued to perform intermittent spot checks of scoring accuracy. This information has been added to the Methods (lines 204-209).

2) For supplemental figure 1, in addition to adding the standard deviation bars, please indicate how many times each stimulus male was used within each housing type.

 We have added errors bars indicating standard errors to Figure S2 (previously Figure S1), and the number of times each visitor male was used within each housing type has been added to the figure legend.

3) Can the authors explain the use of Tukey’s HSD for your post-hoc analysis? This test is not very conservative, and thus seems unlikely to control for spurious significance in the results.

The Tukey HSD test is a common and well-accepted post-hoc test that is appropriate for our analyses. To re-assure the Reviewer that our results have not been unduly influenced by our choice of post-hoc test, we repeated our post-hoc pairwise comparisons of time spent in visitor-initiated aggressive behavior, this time using the Bonferroni-adjusted post-hoc tests. With this analysis, we still find that time spent in visitor-initiated aggressive behavior is significantly greater for MM pairs that contained a single-housed resident than for MM pairs that contained a group-housed resident (p = 0.01).

4) In the discussion, the authors posit that (line 335), “perhaps [single-housed males] prolonged altercations.” While the data you provide is shown as total time, I assume it can be broken down to characterize the lengths of individual fights or aggressive behaviors? Can this be used to provide evidence for/against this possibility?

We include this analysis in response to major point #8 above and in the Results section (lines 275-279). We have modified the relevant sentence within the Discussion to better reflect these additional findings (lines 367-368).

5) Figure 3D: the “MF>MM” seems to just be floating here. That could easily be interpreted that non-huddling is greater in MF than in MM. I would suggest finding another way to represent this result or at least describe this in the figure caption.

Thanks for noting this potential source of confusion. We have modified the way in which statistical significance is indicated in Figure 3D (now indicated with line + asterisk).

Responses to Reviewer 2 Concerns

1) Lines 111-112: For the phrase “Previous studies have found that rates of rodent USVs

112 are responsive to both short- and long-term social isolation [38,59–62]” please provide the direction of the effect on USVs (increase vs decrease).

Two of the cited studies (refs #38 and #62) report an increase in USV rates following social isolation; one study reports an increase in USV rates, as well as changes in acoustic features (ref #59); one study reports a decrease in USV rates (#61); and one study reports only changes in USV acoustic features but not in rates (ref #60). We note that these studies were heterogeneous in the both the timing and duration of social isolation employed, as well as the sex of the subject animals.

Given that the results of these studies are not uniform in direction, we cannot concisely summarize each of the results but instead have modified this sentence to better reflect that both rates and acoustic features of USVs can be altered by social isolation (line 116).

2) The “visitors” used to intrude on the resident’s home cage were used a number of times such that “no visitor was used more than 11 times within 60 days.” This seems like a large number and it is possible that the experience of the visitors could have influenced the behavior of the residents (as occurred in the current study). Identity of intruders was controlled for, but not the number of times that visitors/intruders were used. The level of stress could vary and perceived defeat could stress the visitor/intruder. Please address this issue.

 Although by and large, three days of social isolation did not affect vole social behaviors, we report that visitor males interacting with single-housed male residents spend significantly more time engaged in visitor-initiated aggression compared to visitor males that interacted with group-housed male residents. The Reviewer raises the important point that repeated social interactions for a given visitor might alter their levels of stress, which in turn might alter the behavior of the visitor and could potentially contribute to this significant result. 

To test this idea, we performed a modified analysis of time spent in visitor-initiated aggressive behavior in MM trials, using a model that controlled for both visitor identity, as well as the number of previous trials in which each visitor was used before the current trial. This modified analysis did not alter our finding that visitor males interacting with single-housed male residents spent more time in visitor-initiated aggression than visitor males interacting with group-housed residents (chi-squared = 1.302, p = 0.007).

3) A visual of the setup for recording the vocalizations would be useful, including the plexiglas sleeve and the placement of the microphone. Please also state the number of animals that were excluded for each of the three reasons stated on 173-175

We now include a photo of the set-up, including a view from the side and an overhead view (Fig. S1).

In addressing the second portion of this comment, we realized that there was a typo in the earlier summary of voles excluded. The total number of excluded trials is 17. Ten trials were excluded because a vole jumped on top of the microphone and/or chamber, 3 trials were excluded because the experimenter stopped the recording before the 30-minute mark, and 4 trials were excluded because the visitor ID was not recorded. This information has been added to the Methods (lines 178-182).

4) In the results it was found that short term isolation of males induced the visitor males to be more aggressive. One reasonable speculative explanation is that isolated males were producing more alarm pheromones. One practical 

---

## [Decision Letter · Decision Letter 1]

21 Oct 2024

Effects of short-term isolation on social behaviors in prairie voles

PONE-D-24-33108R1

Dear Dr. Tschida,

We’re pleased to inform you that your manuscript has been judged scientifically suitable for publication and will be formally accepted for publication once it meets all outstanding technical requirements.

Kind regards,

Wolfgang Blenau

Academic Editor

PLOS ONE

Additional Editor Comments (optional):

Reviewers' comments:

Reviewer's Responses to Questions

**Comments to the Author**

1. If the authors have adequately addressed your comments raised in a previous round of review and you feel that this manuscript is now acceptable for publication, you may indicate that here to bypass the “Comments to the Author” section, enter your conflict of interest statement in the “Confidential to Editor” section, and submit your "Accept" recommendation.

Reviewer #1: All comments have been addressed

Reviewer #2: All comments have been addressed

2. Is the manuscript technically sound, and do the data support the conclusions?

Reviewer #1: Yes

Reviewer #2: Yes

3. Has the statistical analysis been performed appropriately and rigorously? 

Reviewer #1: Yes

Reviewer #2: Yes

4. Have the authors made all data underlying the findings in their manuscript fully available?

Reviewer #1: Yes

Reviewer #2: Yes

5. Is the manuscript presented in an intelligible fashion and written in standard English?

Reviewer #1: Yes

Reviewer #2: Yes

6. Review Comments to the Author

Reviewer #1: (No Response)

Reviewer #2: (No Response)

7. PLOS authors have the option to publish the peer review history of their article (what does this mean?). If published, this will include your full peer review and any attached files.

Reviewer #1: No

Reviewer #2: No

---

## [Editor Report · Acceptance letter]

30 Oct 2024

PONE-D-24-33108R1 

PLOS ONE

Dear Dr. Tschida, 

I'm pleased to inform you that your manuscript has been deemed suitable for publication in PLOS ONE. Congratulations! Your manuscript is now being handed over to our production team.

Kind regards, 

on behalf of

Dr. Wolfgang Blenau 

Academic Editor

PLOS ONE